# Holistic engineering of cell-free systems through proteome-reprogramming synthetic circuits

Luis E. Contreras-Llano [1,4], Conary Meyer[1,4], Yao Liu [1], Mridul Sarker[2], Sierin Lim [2], Marjorie L. Longo [3] & Cheemeng Tan [1✉]

Synthetic biology has focused on engineering genetic modules that operate orthogonally from the host cells. A synthetic biological module, however, can be designed to reprogram the host proteome, which in turn enhances the function of the synthetic module. Here, we apply this holistic synthetic biology concept to the engineering of cell-free systems by exploiting the crosstalk between metabolic networks in cells, leading to a protein environment more favorable for protein synthesis. Specifically, we show that local modules expressing translation machinery can reprogram the bacterial proteome, changing the expression levels of more than 700 proteins. The resultant feedback generates a cell-free system that can synthesize fluorescent reporters, protein nanocages, and the gene-editing nuclease Cas9, with up to 5-fold higher expression level than classical cell-free systems. Our work demonstrates a holistic approach that integrates synthetic and systems biology concepts to achieve outcomes not possible by only local, orthogonal circuits.

[1] Department of Biomedical Engineering, University of California, Davis, Davis, CA 95616, USA. [2] School of Chemical and Biomedical Engineering, Nanyang Technological University, 70 Nanyang Drive, Block N1.3, Singapore 637457, Singapore. [3] Department of Chemical Engineering, University of California, Davis, Davis, CA 95616, USA. [4] These authors contributed equally: Luis E. Contreras-Llano, Conary Meyer. ✉email: cmtan@ucdavis.edu

The mantra of synthetic biology advocates for the use of orthogonal genetic modules to engineer and control cellular behavior[1]. However, the use of orthogonal genetic modules often faces the challenges of varying cellular context, such as growth rate, crosstalk, and noise[2]. These challenges highlight the necessity to complement orthogonal genetic module design with a system-based approach that functions in conjunction with cell physiology[3]. Such systems-synthetic biology approaches have been applied in two major ways. First, systems-level properties can be considered for the control of local synthetic modules. For instance, previous studies have investigated the impact of global physiology on cell-free protein synthesis (CFPS)[4]. Second, the global host circuits are modified before the insertion of local synthetic modules. One classic example is the gene knockout and knockin of the BL21 *Escherichia coli* strain for subsequent conversion into BL21 (DE3) using *lac*UV5-T7 RNAP-based synthetic modules[5]. A powerful alternative, referred to as the holistic synthetic biology approach in this work, is to use local synthetic modules to reprogram the global host physiology, which in turn becomes beneficial to the function of the local synthetic modules.

Here, we apply the holistic synthetic biology approach to the engineering of CFPS systems. CFPS decouples cellular growth from protein production, allowing for applications such as synthesis of toxic or metabolically interfering proteins[6,7] paper-based diagnostics[8,9], a priori prediction of metabolic burden[10] and function of genetic circuits[11,12], high-throughput screening[13], and construction of artificial cells[14]. To produce CFPS, cells are grown to mid-exponential phase and then lysed to produce the whole-cell extract. The resulting cell lysate is then used in CFPS reactions by supplementing it with salts, energy sources, amino acids, and dNTPs. Attempts to improve CFPS have focused on the deletion of proteins that drain resources from CFPS, including nucleases[15,16], proteases[17,18], and enzymes involved in amino acid metabolism[19,20]. Purified proteins, such as molecular chaperones[21–24], transcription, and translation machinery[25,26], have also been added to CFPS reactions. In addition, genome recoding approaches have been used to modify the proteome of source bacteria through gene knockins[27] or knockouts[28]. While these approaches can precisely change the concentration of a few proteins, they are challenging to scale up for targeting multiple pathways that can impact CFPS. Furthermore, manipulating the expression levels of many essential genes while maintaining cell viability is often inhibitively complex. A holistic synthetic biology approach could overcome this challenge.

## Results

### Enhanced CFPS via overexpression of translation machinery.
To implement the holistic approach, we used synthetic modules to express all or a subset of the 34 proteins of the core *E. coli* translation machinery within multiple strains of *E. coli* BL21 (DE3) that were lysed and used in CPFS. Distribution of protein overexpression across multiple strains was chosen to decrease the metabolic burden caused by protein expression and plasmid maintenance. The burden imposed by plasmid maintenance manifests in the form of decreased growth rates[29], which in turn generates lower concentrations of ribosomes and other translation machinery proteins[4]. This has been shown to be a limiting factor for efficient CFPS[25]. We hypothesized that the overexpression of translation machinery should benefit CFPS in two ways. First, it should compensate for the increased metabolic burden by virtue of being supplied with translation factors. And second, it should shift the global proteome to a high-growth-rate-like state where translation factors are enriched, and the cell reaches peak protein synthesis efficiency. We produced two different microbial consortia, one with 18 strains (BL-18S) and the

other with 7 strains (BL-7S) to obtain cell lysates enriched in translation machinery without the need to purify and supplement individual proteins. BL-18S expressed 11 initiation, elongation, and termination factors (IETs), as well as 23 aminoacyl-tRNA transferases (AAT). BL-7S expressed 11 IETs and 1 AAT (Fig. 1a and Supplementary Table 1). Throughout this study, we used the expression level of deGFP, a truncated version of eGFP with the same fluorescence properties[30] to quantify the absolute yield of the CFPS (Supplementary Fig. 6, "Methods, Section M4", and Supplementary Note 1). In addition, we optimized the reaction buffer and lysate preparation for the new CFPS (Supplementary Figs. 1–4 and Supplementary Note 2).

To compare our modified extracts to existing systems, we ran several experiments to quantify the differences. The whole-cell lysate of BL-18S (BL-18S$_{WCE}$) and BL-7S (BL-7S$_{WCE}$) had comparable expression activities (Supplementary Fig. 5). Thus, we proceeded with BL-7S$_{WCE}$ due to its simpler preparation procedure. To assess the influence of translation machinery overexpression in CFPS, we compared the protein yield against a cell lysate produced using *E. coli* BL21(DE3) without any plasmids (BL-E$_{WCE}$), a cell lysate from the same strain carrying the original plasmid vectors (BL-P$_{WCE}$), and the commercial S30 T7 High-Yield Expression System (Promega Corporation) (S30) ("Methods, Section M2"). In batch reaction mode, BL-7S$_{WCE}$ produced a maximum of 1.51 mg mL$^{-1}$ of deGFP (Fig. 1b), and 4.8 mg mL$^{-1}$ in a semicontinuous exchange mode (Fig. 1c). The S30 expression system performed poorly when adapted to the semicontinuous exchange mode. Therefore, the data were not included because the protocol for this setup was not defined by the manufacturer. The yields of BL-7S$_{WCE}$ were two- to threefold higher than the controls in both formats. When examining the expression dynamics (Fig. 1d), CFPS reactions assembled using our in-lab cell lysates show a 20 min lag-period before the production of deGFP can be detected. During this initial lag-period, the transcription machinery (T7 RNAP) likely ramped up mRNA synthesis until the mRNA reached the concentration necessary for starting protein synthesis. Once protein synthesis was started, reactions assembled using BL-7S$_{WCE}$ expressed deGFP at a higher rate than other cell lysates. Altogether, the data show that BL-7S$_{WCE}$ can achieve higher expression levels than conventional systems.

### Effect of translation machinery concentration on CFPS.
Our next set of experiments are intended to investigate the cause of the improved CFPS efficiency. Specifically, we sought to decouple the direct effect of increasing the translation factor concentrations in the CFPS reaction from the indirect effects of protein overexpression and feedback from the overexpressed proteins. To study the effects of the increased translation factors in a standard CFPS reaction, we purified the translation machinery proteins overexpressed in BL-18S and supplemented it to BL-E$_{WCE}$ (Fig. 2a, "Methods, Sections M3 and M6"). The expression level of deGFP increased proportionally with the addition of translation machinery (Fig. 2b). These results demonstrate that the increased concentration of protein machinery is not the only factor responsible for the increased protein expression of our multi-strain CFPS systems. Furthermore, we intended to rule out any additional effects that plasmid maintenance or protein overexpression could be causing in our multi-strain CFPS systems. To rule out the plausible effects, we purified translation machinery proteins (overexpressed in BL-7S) and supplemented them to an extract that was generated from BL21 (DE3) overexpressing cyan fluorescent protein (CFP) (BL-CFP$_{WCE}$; "Methods, Sections M2, M3, and M6"). The expression level once again increased proportionally with the amount of protein added but plateaued at a twofold increase (Fig. 2c). Our results show that

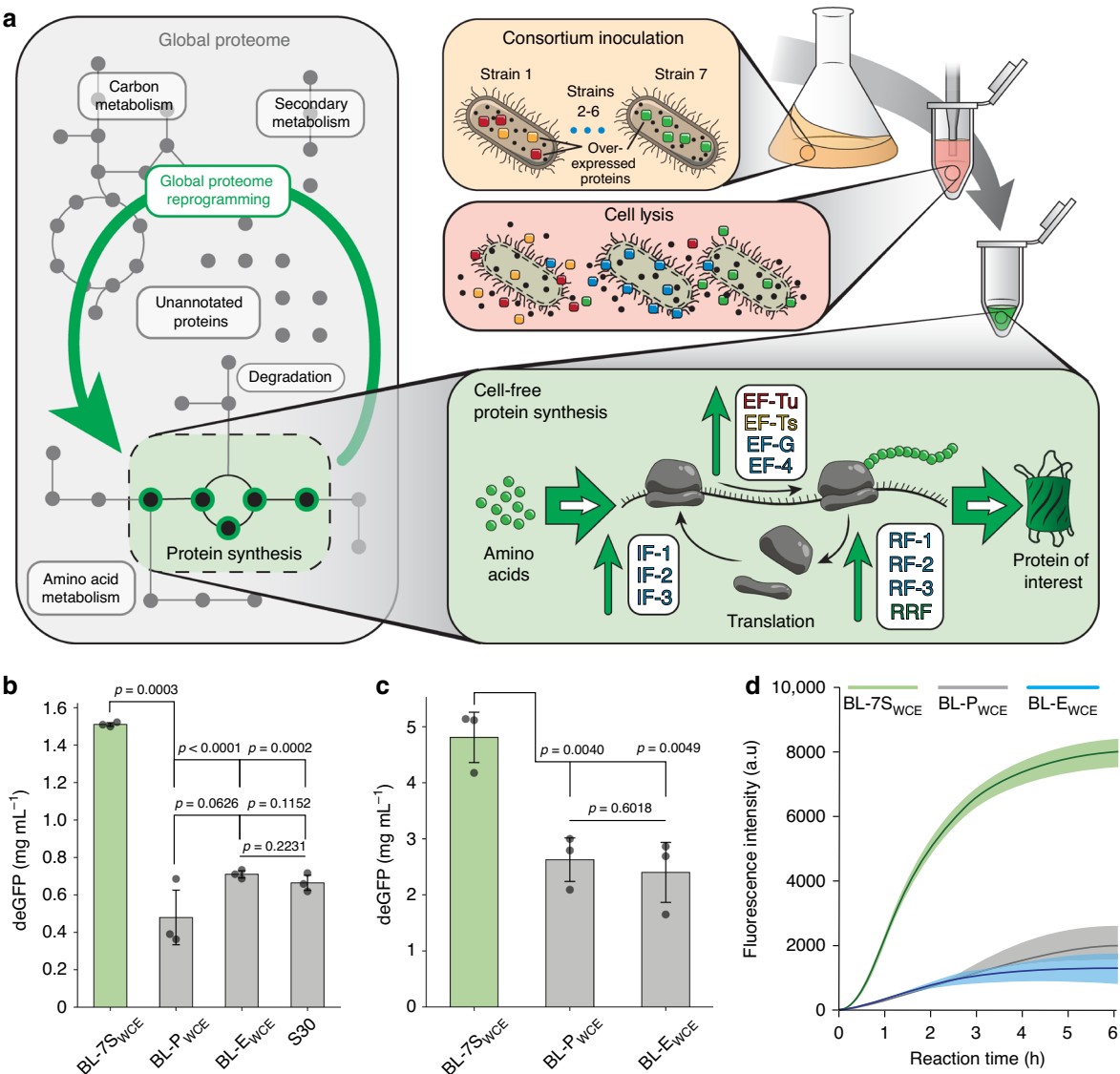

**Fig. 1 A holistic synthetic biology approach to enhance cell-free systems. a** Graphical representation of the production of BL-7S$_{WCE}$ showing the overexpression of 11 translation factors and their influence in different stages of translation. **b** BL-7S$_{WCE}$ exhibits a higher yield than conventional cell lysates in batch CFPS reactions. We expressed deGFP encoded in the plasmid pIVEX-Eps-deGFP (10 µg µL$^{-1}$) using BL-7S$_{WCE}$, BL-E$_{WCE}$, BL-P$_{WCE}$, and the commercial S30 System. Reaction assembly, incubation conditions, and deGFP quantification are described in "Methods, Sections M3 and M4". Reactions assembled using BL-7S$_{WCE}$ exhibit deGFP expression approximately two- to threefold higher than the controls. Data are presented as mean values and error bars represent s.d. ($n = 3$ independent experiments). Standard two-tail $t$-test. **c** BL-7S$_{WCE}$ exhibit approximately twofold more deGFP expression than the controls in semicontinuous exchange reactions. Assembly of the reactions and deGFP quantification are described in "Methods, Sections M4 and M5". Data are presented as mean values and error bars represent s.d. ($n = 3$ independent experiments). Standard two-tail $t$-test. **d** Time series showing the expression dynamics of BL-7S$_{WCE}$ compared with controls under semicontinuous agitation. Reaction assembly, incubation conditions, and deGFP quantification are described in "Methods, Sections M3 and M4". Data are presented as mean values and error bars represent 95% confidence interval ($n = 4$ independent experiments). Source data for **b**–**d** are provided as a Source Data file.

while the concentrations of translation machinery are comparable between single-strain preparations supplemented with purified translation machinery and multi-strain preparations (Supplementary Note 3 and Supplementary Fig. 7), the yields obtained in CFPS are not equivalent. These results are consistent with our hypothesis that the overexpression of translation machinery causes an auxiliary effect on the host circuits that create an environment more favorable for CFPS.

**Influence of protein overexpression on the host proteome.** The above results suggest that the protein profile in our BL-7S$_{WCE}$ is more amenable to CFPS than any of the controls. Specifically, our

data indicate that this proteome reprogramming occurs directly as a result of the overexpression of translation machinery by our synthetic modules. To further understand the favorable changes occurring in BL-7S, we analyzed the protein composition of several whole-cell lysates through mass spectrometry ("Methods, Section M7"). For this experiment, an additional extract using one of the IET strains was created, specifically Strain-1 that overexpresses the elongation factors EF-Tu and EF-Ts (BL-1S$_{WCE}$). We decided to analyze Strain-1 of our 7-strain consortium due to the major roles of EF-Tu and EF-Ts for increasing elongation rates in CFPS[25,26], and because it represents 50% of the inoculation mixture (Supplementary Table 1). All four

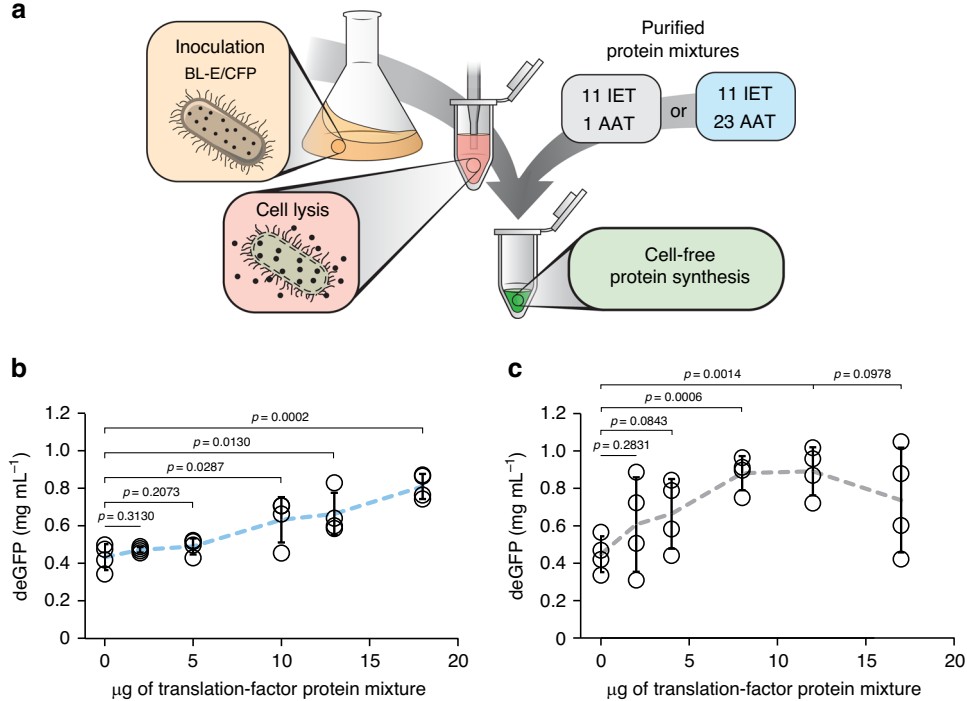

**Fig. 2 Supplemented translation machinery partially enhances protein expression. a** Graphical representation of the supplementation of purified translation factors to CFPS reactions assembled using BL-E$_{WCE}$ and BL-CFP$_{WCE}$. Two different mixtures of purified translation machinery were produced, one with 11 IET (initiation, elongation, termination factors) and 1 AAT (aminoacyl-tRNA transferases) and the other with 11 IET and 23 AAT proteins. **b** Expression yield of BL-E$_{WCE}$ increases proportionally with the supplementation of purified translation machinery. The addition of 10 and 13 µg of purified translation factors to a 10 µL CFPS reaction increases protein yield by ~1.5-fold. Supplementation of up to 18 µg of purified translation factors increases protein yield by approximately twofold. Reaction assembly, purified protein supplementation, and incubation conditions are described in "Methods, Sections M3 and M6". Data are presented as mean values and error bars represent s.d. ($n = 4$ independent experiments). Standard two-tail $t$-test. **c** Expression yield of BL-CFP$_{WCE}$ increases proportionally with the supplementation of purified translation machinery. The addition of 8 and 12 µg of purified translation factors to a 10 µL CFPS reaction increases protein yield by approximately twofold. Supplementation of up to 17 µg of purified protein does not result in any further increase. Reaction assembly, purified protein supplementation, and incubation conditions are described in "Methods, Sections M3 and M6". Data are presented as mean values and error bars represent s.d. ($n = 4$ independent experiments). Standard two-tail $t$-test. Source data for **b**, **c** are provided as a Source Data file.

extracts (BL-7S$_{WCE}$, BL-1S$_{WCE}$, BL-CFP$_{WCE}$, and BL-E$_{WCE}$) were digested, labeled, and subjected to tandem mass tag (TMT) mass spectrometry in quadruplicate ("Methods, Section M7"). The expression capacity of these four extracts agreed with previous results (Fig. 3a). The mass spectrometry data revealed the levels of ~2000 different *E. coli* proteins in all samples. After internal reference scaling ("Methods, Section M7", Supplementary Fig. 8), the data were analyzed by principal component analysis (PCA), showing clear clustering of replicates and separation of experimental conditions (Fig. 3b). The results also show the expected enrichment of overexpressed CFP and translational machinery (Fig. 3c and Supplementary Note 4).

BL-7S indeed showed a global difference in protein content compared with the controls. The proteome of BL-E and BL-CFP were clustered separately in the PCA (Fig. 3b and Supplementary Fig. 9), while the clusters of BL-7S and BL-1S overlapped partly. The overlap between the proteome of BL-7S and BL-1S was anticipated because Strain-1 makes up a majority of BL-7S. To investigate the proteome changes that underlie the clustering, we plotted the fold change of each protein intensity compared with BL-E and the $p$ value from a two-way $t$-test of that comparison (Fig. 3c). On the one hand, the proteome of BL-CFP remained mostly unchanged with a nearly even split between the number of up- and downregulated proteins (changes >25%, $p < 0.01$). On the other hand, the proteome of BL-7S and BL-1S showed a decrease in over a third of all observed proteins, while <5% of all proteins were upregulated. Even though BL-CFP showed a proteome shift,

likely caused by the metabolic burden of protein overexpression, the proteome change did not boost the yield of BL-CFP$_{WCE}$. These results show that the proteome of BL-7S was affected by the expression of the translational machinery. This change in protein profile and content likely results in the generation of an environment more favorable for CFPS.

We further characterized the proteome change uncovered by our mass spectrometry results. To this end, we categorized each protein based on their assigned gene ontological function ("Methods, Section M7"). We then summed the intensity of each protein in each category for protein content comparison (Fig. 3d). Again, the proteome of BL-CFP and BL-E exhibited no significant difference. However, BL-7S and BL-1S exhibited a 17% increase in the gene expression category (e.g., translation factors, aminoacyl tRNA synthetases, ribosomes). They also showed a decrease of 14% in the metabolism (e.g., TCA cycle and amino acid catabolism) and 3% in the homeostasis (e.g., iron homeostasis, proteases, and cell cycle regulators) categories. To better understand the specific proteome changes, more detailed functions were assigned to the proteins. The fold changes between the means of each protein in each extract were compared with the BL-E control. The proteins were then grouped by their function, and the average of all the fold changes was calculated (Fig. 3e). The results show that the expression of translation machinery from a local genetic module results in a global proteome shift generally associated with a cellular state at high-growth rates[4,29]: upregulation of proteins involved in macromolecule synthesis (e.g., chaperones and

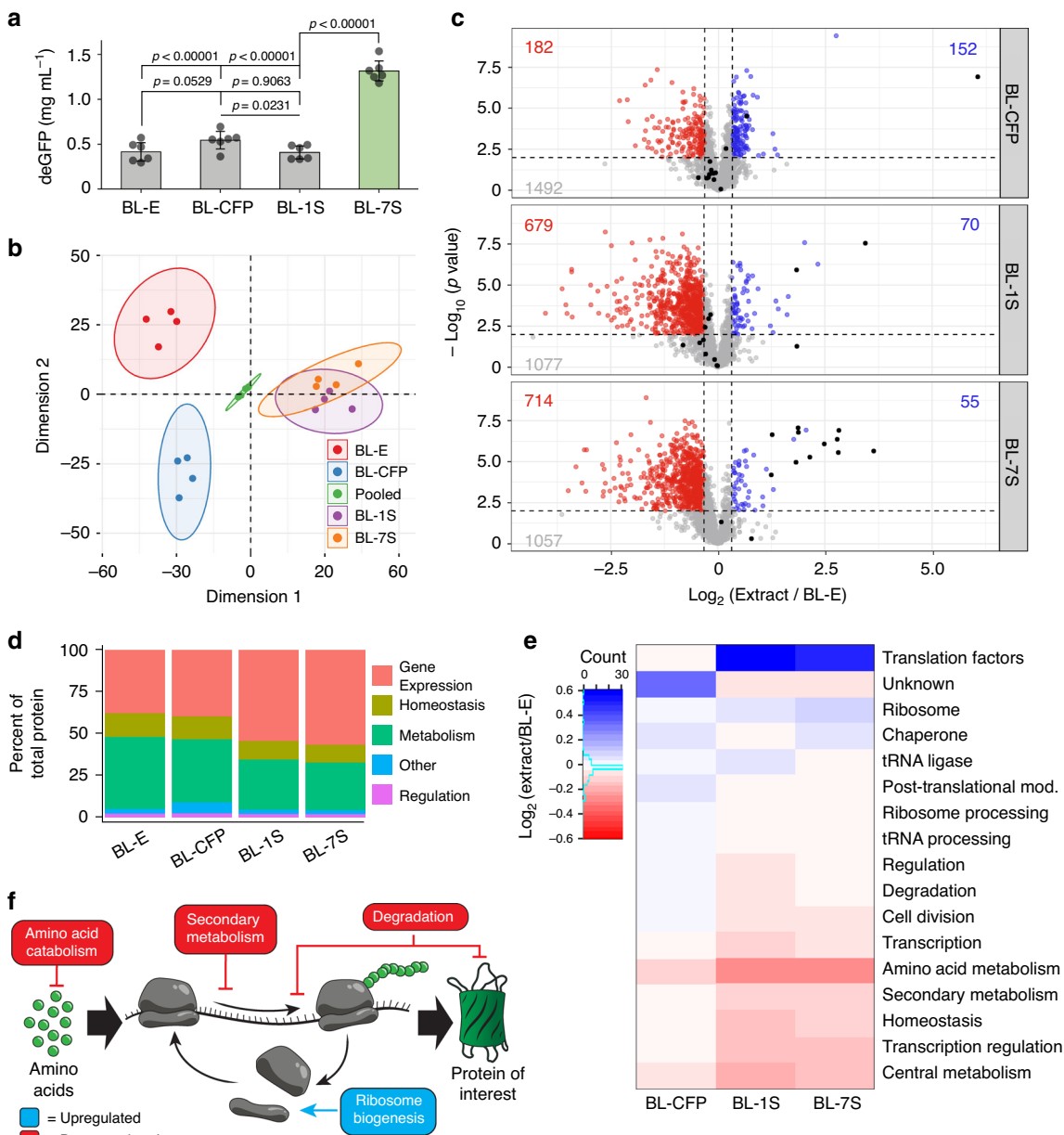

**Fig. 3 Mass spectrometry reveals proteome reprogramming. a** Expression data of all extracts used in mass spectrometry analysis. Data are presented as mean values and error bars represent s.d. ($n = 6$ three independent reactions of two independent extract preparations). Standard two-tail $t$-test. **b** Principal component analysis showing the grouping of the various extracts based on their protein profile. Ellipses indicate the boundary for statistical significance by a two-tailed $t$-test ($p$ value < 0.01). Clear grouping is observed between the replicates of each sample. **c** Volcano plots displaying the changes in protein intensity between each extract and the BL21(DE3) control. Red data points indicate a decrease in protein intensity greater than 25% and a $p$ value < 0.01 from a two-sided $t$-test. Blue data points indicate an increase in protein intensity greater than 25% and a $p$ value < 0.01 form a two-tailed $t$-test. The proteins that were intentionally overexpressed are colored black. Numerous statistically significant changes are observed, many of which are downregulated. **d** The sum of protein intensity in each category for each extract is presented as percentages of the total protein intensity in that extract. There is a marked increase in gene expression and a decrease in metabolism and homeostasis related proteins when comparing BL-1S and BL-7S to BL-E. **e** The identified proteins were further subdivided into more specific functional groups. The log difference between the protein intensity of each extract and BL-E were calculated and then averaged within each functional group and plotted on a colorimetric scale. The dendrogram represents the clustering of each protein group based on their similarity to other groups. CFP was inserted into the Unknown category. **f** Diagram of translation indicating the key changes in the proteome and their influence on gene expression. Source data for **a**–**e** are provided as a Source Data file. In **b**–**e** use the same samples throughout. ($n = 4$ two independent samples from two independent extract preparations).

ribosomal proteins); and downregulation of metabolic proteins that compete with nutrients in CFPS (e.g., tryptophanase and pyruvate kinase) (Fig. 3f). However, we note some exceptions to this general expectation, such as increases in a few metabolic proteins, including glycerol-3-phosphate acyltransferase and 2,3-dihydroxybenzoate-AMP ligase (Supplementary Note 4).

**Demonstrating the versatility of enhanced CFPS**. To explore the potential of our BL-7S$_{WCE}$ beyond the enhanced deGFP expression (Fig. 1b), we decided to test its versatility through different applications. For our first trial, we produced ferritin from *Archaeoglobus fulgidus* (AfFtn), an archaeal iron storage protein capable of self-assembly forming nanocages. AfFtn has been

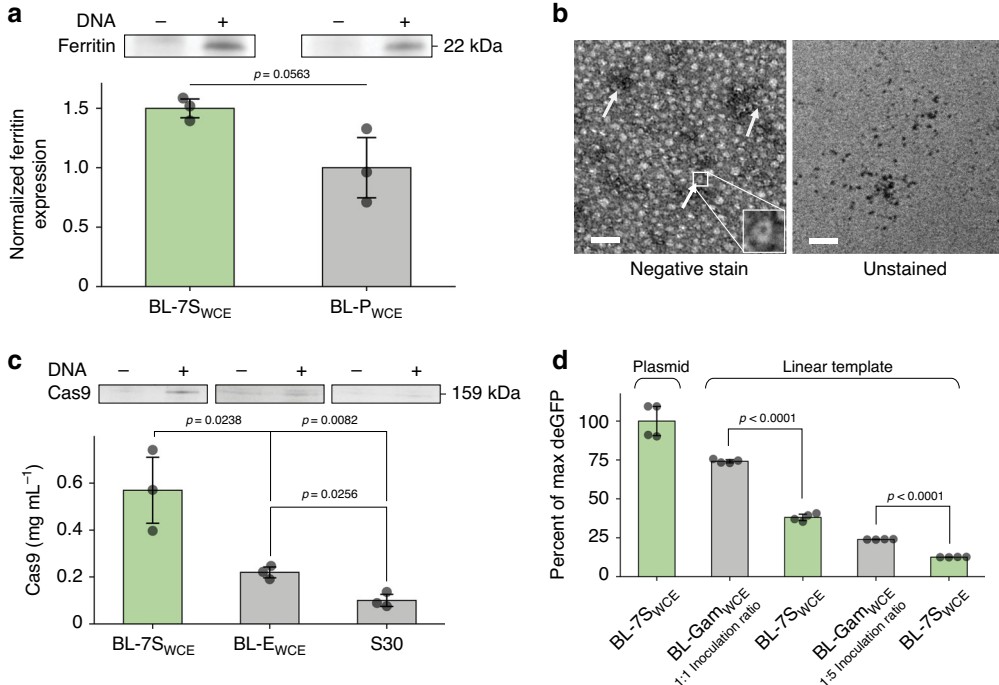

**Fig. 4 Enhanced CFPS as a versatile tool for the expression of diverse proteins. a** CFPS of ferritin using BL-7S$_{WCE}$ and BL-P$_{WCE}$. The bar chart shows that the expression of ferritin in reactions assembled using BL-7S$_{WCE}$ is ~0.5-fold higher than the expression achieved using BL-P$_{WCE}$. Standard two-tail $t$-test ($n$ = 3 independent experiments). Top panel: representative SDS-PAGE results of CFPS reactions expressing ferritin (+) and negative control without plasmid (−). See Supplementary Fig. 10. for the SDS-PAGE analysis of all the CFPS reactions. **b** TEM images of ferritin nanocages. The left image shows the stained samples, while the right image shows the unstained samples. In the unstained image, the iron core of the ferritin cages can be seen. White arrows indicate nanocages. See Supplementary Fig. 10 for a side by side comparison with negative controls. Three independent experiments of the assembly of ferritin nanocages and its imaging using TEM (See "Methods, Section M9") were performed. All experiments showed the same results. Scale bar represents 100 nm. **c** CFPS of Cas9 using BL-7S$_{WCE}$ and control extracts. The bar chart shows that the expression of Cas9 in reactions assembled using BL-7S$_{WCE}$ is approximately three- and fivefold higher than the expression achieved using BL-E$_{WCE}$ and S30, respectively. See "Methods, Section M8" for details about Cas9 quantification. Standard two-tail $t$-test ($n$ = 3 independent experiments). Top panel: representative SDS-PAGE results of CFPS reactions expressing Cas9 (+) and negative control without plasmid (−). See Supplementary Fig. 11. for the SDS-PAGE analysis of all the CFPS reactions. **d** CFPS of deGFP using linear DNA as a template. The reactions assembled using 1:1 and 1:5 inoculation ratios of BL-Gam$_{WCE}$ and linear DNA as a template show 74% and 24% of the maximum deGFP expression respectively compared with a control assembled using BL-7S$_{WCE}$ and plasmid DNA as a template. These reactions exhibit up to approximately twofold higher deGFP expression than controls assembled using BL-7S$_{WCE}$ and linear DNA as a template. Standard two-tail $t$-test ($n$ = 4 independent experiments). In **a**, **c**, **d** data are presented as mean values and error bars represent s.d. Source data for **a**, **c**, **e** are provided as a Source Data file.

shown to encapsulate and release molecular cargo[31]. As AfFtn requires the assembly of precisely 24 subunits of 22 kDa to form nanocages, it is a good test case for the CFPS system to produce large protein assemblies while maintaining its function. Reactions assembled with BL-7S$_{WCE}$ expressed 50% more ferritin than our controls assembled with BL-E$_{WCE}$ (Fig. 4a and Supplementary Fig. 10A, B). TEM images demonstrate the AfFtn nanocages of 12 nm (Fig. 4b and Supplementary Fig. 10C). The iron core formation in the unstained TEM images confirms the function of the produced AfFtn.

One of the major challenges of *E. coli* based CFPS systems is their limited ability to efficiently synthesize large proteins. This problem becomes particularly pronounced in the expression of proteins larger than 70 kDa[32]. We decided to test if our multi-strain system offers an advantage over traditional approaches in this task. Thus, we expressed the biotechnologically relevant protein Cas9 (159 kDa) and compared its expression against BL-E$_{WCE}$ and S30 (Fig. 4c and Supplementary Fig. 11). Our BL-7S$_{WCE}$ produced 0.52 mg mL$^{-1}$ of Cas9. This is approximately threefold higher than BL-E$_{WCE}$ and approximately fivefold higher than S30. These results show that our system can synthesize broad sizes of proteins between 20 and 160 kDa in higher quantities than conventional systems.

The modularity of the bacterial consortium enables the incorporation of additional strains in our system as a plug-and-play feature. By exploiting this feature, we could confer a new function to our cell-free system, such as expressing proteins from linear templates. To implement this, we added a strain expressing the Gam protein (a nuclease inhibitor), resulting in an 8-strain WCE (BL-8S-Gam$_{WCE}$; "Methods, Section M2"). We used rolling circle amplification (RCA) to generate the deGFP encoding template ("Methods, Section M3"). The resulting double-stranded linear DNA was added directly (21% V/V) into three different cell lysates: two BL-8S-Gam$_{WCE}$ with different inoculation ratios of the Gam-expressing strain (1:1 and 1:5), and controls without Gam (BL-7S$_{WCE}$). Using the amplified linear DNA, the two BL-8S-Gam$_{WCE}$ synthesized approximately twofold more deGFP than the BL-7S$_{WCE}$ controls (Fig. 4d). The deGFP expression levels increased proportionally with the amount of Gam-expressing strain. The maximum yield achieved using the linear template in reactions assembled with BL-8S-Gam$_{WCE}$ is ~75% of the yield achieved using BL-7S$_{WCE}$ and plasmid template (Fig. 4c). The results of this experiment show how the modularity of our bacterial consortium approach can be exploited to custom modify cell lysates to match the requirements of a given experiment. However, further optimization and benchmarking may be

necessary to make a fair comparison between our and commercial CFPS systems that are designed for specific applications. Altogether, these applications demonstrate the power of our holistic synthetic biology approach in generating versatile high-yield CFPS systems.

## Discussion

Our work highlights both the utility and the potential of holistic synthetic biology approaches in boosting the performance of local synthetic modules. We demonstrate that the proteome repro-graming described in our study is the direct result of the over-expression of translation machinery in the host cells. Furthermore, we show that the use of a plasmid system in the source strain does not result in a decrease in the activity of the CFPS system. This illustrates how plasmid-based approaches can be implemented to functionalize cell lysates without sacrificing CFPS efficiency. This study opens a new research direction in cell-free synthetic biology, showcasing how the integration of orthogonal circuits, cell physiology, and systems biology can become a powerful tool that maximizes the output of a given cell-free system. Similar approaches have been used for transcriptional rewiring with the aim of increasing the production of proteins and metabolites in vivo[33,34]. However, in order to refine holistic and transcriptional rewiring approaches, there are still many challenges ahead. For instance, precise molecular details of the feedback loop generated by these approaches are not fully eluci-dated. Understanding precisely how these positive feedback loops work could allow precise control over the targeted metabolic pathways, and tight regulation of individual protein levels. If this is achieved, the possibilities for the application of such a holistic approach are vast, ranging from the engineering of mammalian cells to the control of disease development. For instance, our holistic approach can be used to accelerate other work on cell-free systems, including the incorporation of non-natural amino acids into proteins[35], posttranslational protein modifications[36], ribo-some engineering[37], and the production of stable and functionally folded membrane proteins[38]. The benefits of exploiting the ben-eficial crosstalk between synthetic modules and host biological programs could open a new era in synthetic biology.

## Methods

**M1: Construction of plasmids and strains**. We used the plasmids pIVEX2.3d (Roche), pET15b (Novagen), pLysS (Novagen), and pSC101[39] as the backbones for all our constructs. The backbones of pET15b, pLysS, and pSC101 were used to create the plasmids pIURAH, pIURCM, and pIURKL, respectively. Briefly, the three plasmids have compatible replication origins, distinct copy number, carry a NsiI/PacI cloning site downstream of a PT7–lacO hybrid promoter, and have a T7 RNAP terminator sequence. pIURAH contains the ampicillin resistance gene/ColE1 replication origin and expresses lacI, pIURCM contains the chloramphenicol resistance gene/p15A replication origin and expresses T7 lysozyme, and pIURKL contains kanamycin resistance gene/pSC101 replication origin. The plasmids pIURAH and pIURKL were used as backbones to generate all 34 vectors encoding translation machinery by cloning the translation machinery genes into one of these plasmids (See Supplementary Table 1). Based on previous literature[40], a 6×-His-tag was also added to each gene at either the N or C terminus to allow for the purification of the translation machinery proteins. The plasmids pIURAH, pIURCM, pIURKL, pET15bL-CFP, and all 34 translation machinery expressing plasmids were made by Villareal et al.[41] and are available through Addgene [https://www.addgene.org/Cheemeng_Tan/]. The construct pIVEX-deGFP was generated by PCR amplifying the sequence of deGFP from the plasmid pBEST-OR2-OR1-Pr-UTR1-deGFP-T500 (Addgene, Cat# 40019)[30] and inserting it into the PCR amplified backbone pIVEX using Gibson Assembly (New England Bio-Labs, Inc). The construct pIVEX-Eps-deGFP was built as described above for the plasmid pIVEX-deGFP, but an additional epsilon sequence (TTAACTTTAA)[42] was inserted between the T7 promoter and the RBS (Supplementary Fig. 6). The construct pIVEX-Eps-Cas9 was generated by PCR amplifying the sequence of Cas9 from the plasmid pwtCas9-bacteria (Addgene, Cat# 44250)[43] and inserting it into the PCR amplified backbone pIVEX-Eps using Gibson Assembly. The plasmid pIURAH-Gam was built by PCR amplifying the sequence of Gam from the plasmid pKDsgRNA-p15 (Addgene, Cat# 62656)[44] and inserting it into the PCR amplified

backbone pIURAH using Gibson Assembly. All resulting plasmids were trans-formed into and propagated using *E. coli* Top-10 cells (Thermo Fisher Scientific).

*E. coli* BL21(DE3) is used throughout this study to build all the strains used to produce all of our cell-free lysates. The 18-translation machinery overexpressing strains were produced by transforming *E. coli* BL21(DE3) with the plasmids specified in Supplementary Table 1, and with the plasmids pIURAH, and pIURKL (only for strain 7) without expression cassettes. Each strain is designed to overexpress one or two translation machinery proteins upon IPTG induction, and all strains have antibiotic resistance to carbenicillin, chloramphenicol, and kanamycin. More details about the design of the strains can be found in our previous work[41]. BL21 (DE3) was transformed with the plasmids pIURAH, pIURCM, and pIURKL without expression cassettes to generate the strain used to produce our control with antibiotic resistance to carbenicillin, chloramphenicol, and kanamycin. Our CFP expressing strain was generated by transforming the plasmids pET15bL-CFP, pIURCM, and pIURKL into BL21 (DE3). Our Gam-expressing strain was generated by transforming the plasmids pIURAH-Gam, pIURCM, and pIURKL into BL21 (DE3).

**M2: Preparation of whole-cell extracts**. For our whole-cell extract preparations, we variate the specific strain or consortium used and the inoculation ratios (ratio represent % of the strain in the total volume of the mix). Culture and induction times and all subsequent steps were made generic among all preparations.

BL-7S$_{WCE}$ and BL-18S$_{WCE}$ were prepared using the following protocol: each strain comprising the 7 or the 18-strain consortium was individually grown in 3 mL of 2YTP media supplemented with carbenicillin/chloramphenicol/kanamycin at 37 °C with shaking at 200 rpm overnight. The overnight cultures were used to establish the BL-7S or BL-18S consortia by mixing strains at the indicated ratios (See Supplementary Table 1). The mixtures were then used to inoculate 300 mL of 2YTP supplemented with carbenicillin and kanamycin at a 1/250 dilution.

BL-E$_{WCE}$ was prepared using the following protocol: the strain BL21 (DE3) was grown in 3 mL of 2YTP media at 37 °C with shaking at 200 rpm overnight. The saturated overnight culture was then used to inoculate 300 mL of 2YTP at a 1/250 dilution.

BL-P$_{WCE}$ was prepared using the following protocol: the strain BL21 (DE3) transformed with the plasmids pIURAH, pIURCM, and pIURKL was grown in 3 mL of 2YTP media supplemented with carbenicillin/chloramphenicol/kanamycin at 37 °C with shaking at 200 rpm overnight. The saturated overnight culture was then used to inoculate 300 mL of 2YTP supplemented with carbenicillin and kanamycin at a 1/250 dilution.

BL-CFP$_{WCE}$ was prepared using the following protocol: the strain BL21 (DE3) transformed with the plasmids pET15bL-CFP, pIURCM, and pIURKL was grown in 3 mL of 2YTP media supplemented with carbenicillin/chloramphenicol/kanamycin at 37 °C with shaking at 200 rpm overnight. The saturated overnight culture was then used to inoculate 300 mL of 2YTP supplemented with carbenicillin and kanamycin at a 1/250 dilution.

BL-8S-Gam$_{WCE}$ was prepared using the following protocol: the strains comprising the 7-strain consortium, and the Gam-expressing strain transformed with the plasmids pIURAH-Gam, pIURCM, and pIURKL were individually grown in 3 mL of 2YTP media supplemented with carbenicillin/chloramphenicol/kanamycin at 37 °C with shaking at 200 rpm overnight. The overnight cultures were used to establish the BL-8S consortium by mixing strains at the indicated ratios (Supplementary Table 1). The mixtures were then used to inoculate 300 mL of 2YTP supplemented with carbenicillin and kanamycin at a 1/250 dilution.

The following steps were used for all whole-cell lysate preparations: the culture was incubated at 30 °C, 250 rpm until the OD reached 0.15. The culture is then induced with 0.5 mM IPTG and grown until an OD of 1.0. After induction, bacteria cells were harvested and washed twice with 20 mL of Buffer A (4000 g, 20 min, 4 °C). Buffer A contains 10 mM Tris-acetate pH 7.6, 14 mM magnesium acetate, and 60 mM potassium gluconate. After the final wash and centrifugation, the pelleted cells were weighed and suspended in 1 mL of Buffer A supplemented with 2 mM DTT (Thermo Fisher Scientific) per 1 g of wet cell mass. To lyse cells by sonication, freshly suspended cells were transferred into 1.5 mL microtubes and placed in an ice-water bath to minimize heat damage during sonication. The cells were lysed using a Q125 Sonicator with a 2 mm diameter probe at a frequency of 20 kHz and 50% amplitude. Sonication was continued for about 27 cycles 10 s ON/10 s OFF. For each 0.5 mL sample, the input energy was ~1000 J. Cell lysates were centrifuged at 12,000 g for 20 min at 4 °C. The supernatant was collected and incubated at 30 °C for 30 min. The resulting WCE was aliquoted and stored at −80 °C.

**M3: Assembly of CFPS reactions**. The assembly of CFPS reactions for batch experiments was carried out as follows: CFPS reactions (10 μL) were assembled in 1.5 mL low protein binding microcentrifuge tubes (Thermo Scientific) by mixing the following components: 1.2 mM each of ATP and GTP; 0.85 mM each of UTP, and CTP (Promega); 34 μg mL$^{-1}$ folinic acid (Sigma-Aldrich); 170 μg mL$^{-1}$ of *E. coli* tRNA mixture from *E. coli* MRE600 (Roche); 2 mM for each of the 20 standard amino acids (Sigma-Aldrich); 0.33 mM NAD (Roche); 0.27 mM CoA (Sigma-Aldrich); 4 mM spermidine (Sigma-Aldrich); 180 mM potassium glutamate (Sigma-Aldrich); 12 mM magnesium glutamate (Sigma-Aldrich); 50 mM HEPES pH 7.6 (Sigma); 67 mM creatine phosphate (Roche); 80 μg mL$^{-1}$ Creatine Kinase (Roche); 0.64 mM cAMP (Sigma-Aldrich); 2% PEG8k (Sigma-Aldrich); 0.2 mg mL$^{-1}$ BSA;

2.7 μL (27% v/v) of cell extract, and 100 ng plasmid DNA. Each CFPS reaction was assembled on ice and incubated overnight at 30 °C with shaking at 300 rpm unless noted otherwise. As individual reagent concentrations were optimized, their optimal value listed above were used for all reactions from that point onward.

The assembly of reactions supplemented with purified translation machinery mixtures was carried out as follows: reactions were assembled as described above and supplemented with varying amounts of purified translation machinery mixtures. For the experiments in Fig. 2b, we supplemented the 34 translation machinery proteins overexpressed in BL-18S ("Methods, Section M6") to a CFPS reaction assembled with BL-E$_{WCE}$. For the experiments in Fig. 2c, we supplemented the 11 translation machinery proteins overexpressed in BL-7S ("Methods, Section M6") to a CFPS reaction assembled with BL-CFP$_{WCE}$. Supplementation of proteins did not affect the final concentration of any of the components in the CFPS reactions. Negative controls were assembled using the same volume of Buffer A than the volume of supplemented translation machinery mixtures ("Methods, Section M6").

The assembly of reactions under semicontinuous agitation was carried out as follows: reactions were scaled up to 15 μL, assembled into 1.5 mL low protein binding microcentrifuge tubes, and transferred to a 384-well plate (Corning). Once all the reactions were loaded into the plate, the wells were sealed with film and the plate was loaded into an m1000Pro Infinite plate reader to measure fluorescence. Reactions were incubated at 30 °C with semicontinuous shaking at 300 rpm (30 s ON, 30 s OFF) for 12 h. Fluorescence was measured every 10 min and followed for 12 h. Note: the yield of all reactions carried out in 384-well plate format under semicontinuous agitation was considerably lower compared with control batch reactions carried out in parallel. This decrease in yield was consistent across all our in-lab cell lysates and points out to agitation as a crucial parameter for achieving high-yield protein expression.

The assembly of CFPS reactions using a linear template was carried out as follows: we amplified 1 ng of the plasmid pIVEX-Eps-deGFP using the commercial kit TempliPhi for RCA according to the manufacturer's instructions (GE Healthcare, UK). The resulting double-stranded linear DNA template was directly added (21% V/V) to CFPS reactions assembled using BL-7S$_{WCE}$ or BL-8S-Gam$_{WCE}$. This is the maximum percentage by volume that we could add to the CFPS reactions without perturbing the concentrations of the rest of the components. The precise concentration of DNA obtained through RCA could not be quantified using a Nanodrop spectrophotometer. This issue is because even in the absence of input DNA, the RCA reaction yields nonspecific products. However, according to the manufacturer's indications, we estimate that the amount of double-stranded linear DNA added to each CFPS reaction is between 150 and 500 ng.

**M4: Quantification of deGFP expression.** Fluorescent measurements were taken of CFPS reactions diluted 1:50 in dilution buffer (50 mM HEPES pH 7.6, 4 mM spermidine, 2% PEG8k, 12 mM magnesium glutamate, 180 mM potassium glutamate, and 0.4 mg mL$^{-1}$ BSA). Active deGFP protein yields were quantified by measuring fluorescence using a NanoQuant plate (Tecan) and an m1000Pro Infinite plate reader. Excitation and emission wavelength used to measure the fluorescence of deGFP were 488 and 507 nm, respectively. deGFP fluorescence units were converted to concentration using a standard curve. The curve was generated using the pure EGFP standard from Biovision. Previous studies have demonstrated that the fluorescence of deGFP and EGFP are the same and are therefore comparable[30]. The EGFP sample was diluted in dilution buffer and measured to generate the standard curve shown in Supplementary Fig. 6.

**M5: Semicontinuous exchange reaction.** The semicontinuous reactions were conducted using the 96-Well Equilibrium dialyzer (MWCO 10 kDa) purchased from Harvard Apparatus (Holliston, MA). Reactions were set up with 20 μL cell-free reactions loaded on one side of the dialyzer with 200 μL of feeding solution on the other. The feed solution has the same composition as the cell-free reaction, except the whole-cell extract was substituted with Buffer A from the whole-cell extract procedure and the DNA was substituted with water. The reaction was incubated at 30 °C with constant rotation at 0.125 Hz on a rotary axis such that the wells were inverted with each rotation. The reaction was incubated for 24 h prior to measurement.

**M6: Co-purification of translation machinery using a co-culture approach.** We used two different microbial consortia (BL-18S and BL-7S) to purify the 12 (11 IETs and 1 AAT) and 34 (11 IETs and 23 AAT) translation machinery proteins exogenously added to single-strain cell lysates. For these 12 and 34 multi-protein purifications, we followed the protocol for the preparation of BL-7S$_{WCE}$ and BL-18S$_{WCE}$ respectively ("Methods, Section M2"), and couple them with the following steps to co-purify the proteins overexpressed by both consortia. After cell-lysis by sonication, we proceeded to clarify the cell lysate by centrifugation at 20,000 g for 20 min at 4 °C. We collected the supernatant and proceeded with the co-purification of the overexpressed proteins in the cell lysate. The following buffers were prepared in advance and stored at 4 °C for no longer than 24 h. Buffer A contains 10 mM Tris-acetate pH 7.6, 14 mM magnesium acetate, and 60 mM potassium gluconate. Buffer B contains 10 mM Tris-acetate pH 7.6, 10 mM magnesium acetate, and 1 M ammonium chloride. Buffer C contains 10 mM

Tris-acetate pH 7.6, 10 mM magnesium acetate, and 500 mM imidazole. The collected supernatant was diluted fivefold and applied to a 1 mL HisTrap FF column (GE Healthcare Life Sciences) previously equilibrated with 10 volumes of Wash Buffer 1 (Buffer B: Buffer C, 97.5:2.5, supplemented with 2 mM DTT). The column was washed with 10 volumes of Wash Buffer 1, followed by a second washing step with 10 volumes of Wash Buffer 2 (Buffer B: Buffer C, 95:5, supplemented with 2 mM DTT). Proteins were eluted using 7 mL of elution buffer (Buffer B: Buffer C, 20:80, supplemented with 2 mM DTT). Eluted proteins were dialyzed at 4 °C using a 3500 kDa MWCO (Thermo Fisher Scientific) cellulose membrane against Buffer A overnight and after a buffer change for 6 additional hours. Dialyzed proteins were then concentrated by reducing the volume 20-fold using an Amicon Ultra-4 Centrifugal Filter Unit with a 3000 kDa MWCO (Millipore Sigma). The resulting co-purified proteins were aliquoted and stored at −80 °C. Protein concentrations of the co-purified proteins were quantified using the Pierce 660 nm Protein Assay (Thermo Fisher Scientific).

**M7: Mass spectrometry.** The following protocol was used for peptide sample preparation: the proteins in the whole-cell extract preparations were quantified using BCA assay (Thermo Scientific). A volume equal to 150 μg of protein was used for S-Trap (PROTIFI) digestion. Digestion followed the S-trap protocol; briefly, the proteins were reduced and alkylated, the buffer concentrations were adjusted to a final concentration of 5% SDS 50 mM TEAB, 12% phosphoric acid was added at a 1:10 ratio with a final concentration of 1.2% and S-trap buffer (100 mM TEAB in 90% MeOH) is added at a 1:7 ratio (V/V ratio). The protein lysate S-trap buffer mixture was then spun through the S-trap column and washed 3 times with S-Trap buffer. Finally, 50 mM TEAB with 6 μg of trypsin (1:25 ratio) is added and the sample is incubated overnight with one addition of 50 mM TEAB with trypsin after 2 h. The following day the digested peptides were released from the S-trap solid support by spinning at 3000 g for 1 min with a series of solutions starting with 50 mM TEAB which is placed on top of the digestion solution, then 5% formic acid followed by 50% acetonitrile with 0.1% formic acid. The solution is then vacuum centrifuged to almost dryness and resuspended in 2% acetonitrile 0.1% tri-flouroacetic acid and subjected to fluorescent peptide quantification (Thermo Scientific).

The following protocol was used for peptide labeling with TMTs and fractionation: two sets of TMT-10plex labels were used to label the sample. The replicates of each extract were split evenly across the two sets and the tags were assigned such that each replicate had a different mass tag to avoid unintentional bias. In total 20 μg of each sample was diluted with 50 mM TEAB to 25 μL per replicate. Two additional samples consisting of 5 μg of protein from each sample included in each TMT-10plex were pooled together to create a reference to account for bias between the two TMT runs. Each sample was labeled with the TMT-10Plex Mass Tag Labeling Kit (Thermo Scientific). Briefly, 20 μL of each TMT label (126-131) was added to each digested peptide sample and incubated for an hour. The reaction was quenched with 1 μl of 5% hydroxylamine and incubated for 15 min. All labeled samples were then mixed and lyophilized to almost dryness. The TMT labeled sample was reconstituted, desalted, and separated into eight fractions by high pH fractionation (Thermo Scientific). One-third of each fraction (~800 ng) was loaded on to the LC-MS/MS for analysis.

The following protocol was used for liquid chromatography and mass spectrometry of the samples: liquid chromatography separation was conducted on a Dionex nano Ultimate 3000 (Thermo Scientific) with a Thermo Easy-Spray source. The digested peptides were reconstituted in 2% acetonitrile/0.1% trifluoroacetic acid and 1 μg in 5 μL of each sample was loaded onto a PepMap 100 Å 3U 75 μm × 20 mm reverse-phase trap where they were desalted online before being separated on a 100 Å 2U 50 μm × 150 mm PepMap EasySpray reverse-phase column. Peptides were eluted using a 120-min gradient of 0.1% formic acid (A) and 80% acetonitrile (B) with a flow rate of 200 nL/min. The separation gradient was run with 2–5% B over 1 min, 5–50% B over 89 min, 50–99% B over 2 min, a 4-min hold at 99% B, and finally 99% B to 2% B held at 2% for 18 min.

The following protocol was used for mass spectra acquisition: mass spectra were collected on a Fusion Lumos mass spectrometer (Thermo Fisher Scientific) in a data-dependent MS3 synchronous precursor selection method. MS1 spectra were acquired in the Orbitrap, 120 K resolution, 50 ms max injection time, $5 \times 10^5$ max injection time. MS2 spectra were acquired in the linear ion trap with a 0.7 Da isolation window, CID fragmentation energy of 35%, turbo scan speed, 50 ms max injection time, $1 \times 10^4$ AGC, and maximum parallelizable time turned on. MS2 ions were isolated in the ion trap and fragmented with an HCD energy of 65%. MS3 spectra were acquired in the orbitrap with a resolution of 50 K and a scan range of 100–500 Da, 105 ms max injection time, and $1 \times 10^5$ AGC.

The following process was followed for peptide and protein identification: identification of peptides and proteins was conducted using the PAW pipeline[45]. In brief, the ProteoWizard toolkit is used to convert the MS scans into intensity values and extract the TMT reporter ion peak heights. The Comet database search engine is then used to identify peptides. The E. coli BL21 (DE3) proteome UP000002032 and a list of known contaminants and expressed protein sequences were used for protein identification. Results are filtered based on a desired false discovery rate using the target decoy method. Identified proteins with sequence coverage of <5% were excluded from the downstream analysis.

The following process was used to scale the two TMT results: the protein intensities from the pooled samples in each 10plex were used to calculate scaling factors that can be applied to the intensity values from each sample in each TMT-10plex, eliminating the bias that results from independent MS runs[46]. The data are presented before and after normalization in Supplementary Fig. 8.

The following process was used for the assignment of gene ontological function: identified proteins were assigned gene ontological functions based on the gene ontology identifiers provided in the *E. coli* BL21 (DE3) proteome UP000002032. The gene ontology identifiers were grouped based on the general functional categories of interest. The specific identifiers used for this assignment are detailed in Supplementary Data 1. Proteins that lacked identifiers or only possessed broad identifiers were classified as Unknown. Several proteins with known functions that lacked identifiers were manually assigned the correct functional group. These manual assignments are also detailed in Supplementary Data 1.

**M8: Protein quantification and SDS-PAGE analysis.** Analysis of proteins by SDS-Polyacrylamide Gel Electrophoresis (PAGE) was carried out by separating proteins from whole-cell lysates and CFPS reactions using 4–20% Mini-PROTEAN TGX precast gels (Bio-Rad). We used Precision Plus Protein Dual Color Standards (10–250 kDa) as a reference standard for molecular weight verification. Protein gels were endpoint stained using PageBlue Protein Staining Solution (Thermo Fisher Scientific) according to the manufacturer instructions. Gels were imaged using a PXi Imaging system (Syngene) and band analysis and protein quantification were carried out using the open-source platform for biological imaging analysis Fiji (http://fiji.sc/cgi-bin/gitweb.cgi/) and the proprietary software GeneTools (Syngene).

**M9: Transmission electron microscopy.** The following protocol was used for the assembly of the ferritin nanocage: 1 mL of Denaturation Buffer (25 mM HEPES, 50 mM NaCl, pH 7.5) was added to the cell-free reaction after expression of ferritin and then heat-treated at 90 °C for 10 min. Ferrous sulfate heptahydrate was added drop by drop to a final concentration of 2.4 mM. The sample is then incubated overnight at 4 °C. Dynamic light scattering and TEM analyses were performed to confirm the cage assembly.

The following protocol was used for TEM sample preparation: samples were adsorbed on to the carbon-coated electron microscopy grid (Formvar carbon film on 300 mesh copper grids, Electron Microscopy Science) and dried at room temperature for 5 min. For samples with negative staining, the grid was placed on a droplet of 1.5% uranyl acetate for 3 min and the excess stain was removed with a soft wipe. The grid was air-dried for 5 min. All grids were stored in a drying cabinet until further use. The images were obtained in a transmission electron microscope (JEOL JEM-1400) operating at 100 kV.

**M10: Statistical analysis of results.** Unless other is specified, statistical tests were performed using a standard two-tailed *t*-test. The exact *p* value for each statistical analysis is reported directly in the figures unless $p < 0.00001$. The number of replicates contributing to the calculation is listed in the figure legends. All error bars and measures of central tendency are defined in the figure legends.

**Reporting summary.** Further information on research design is available in the Nature Research Reporting Summary linked to this article.

## Data availability

All data used to generate Figs. 1b–d, 2b, c, 3a–e, 4a, c, d, and Supplementary Figs. 1A–D, 2A–D, 3, 4, 5, 6, 8, and 9 in this paper are included as a Source Data File. The identified protein abundances and internal reference normalized data from the proteomics study are provided in Supplementary Data 1. This file also includes the specifications for gene ontological assignments of each protein. The mass spectrometry proteomics data have been deposited to the ProteomeXchange Consortium via the PRIDE[46,47] partner repository with the dataset identifier PXD018858 [https://www.ebi.ac.uk/pride/archive/projects/PXD018858]. The plasmids used to make the strains included in this study are available at [https://www.addgene.org/Cheemeng_Tan/]. Any other relevant data are available from the authors upon reasonable request.

## Code availability

All custom code used to interpret and analyze the protein abundances was deposited in GitHub and its publicly available at [https://github.com/ccmeyer/TMT-analysis].

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

## Acknowledgements

We thank Michelle Salemi and Brett Phinney from the UC Davis Proteomics core facility for their technical assistance with the mass spectrometry sample preparation and processing. We thank Rupali Reddy Pasula for technical assistance in performing and analyzing the TEM imaging of ferritin. We appreciate the discussion of the paper with the members of the Tan lab. This work is supported by a UC-MEXUS Doctoral Fellowship to L.E.C-L., and partially supported by the National Science Foundation under Grant No. (1806366 and 1808237). The mass spectrometry experiments were funded by the UC Davis Campus Research Core Facilities (CRCF) grant 1S10OD021801-01.

## Author Contributions

L.E.C-L., C.M., and C.T. wrote the paper. L.E.C-L., C.M., and C.T. conceived the work. L.E.C-L. and C.M. performed all experiments. C.M. analyzed the mass spectrometry data. L.E.C-L. performed protein purification. Y.L. performed reaction buffer and protocol optimization. M.S. and S.L. performed the TEM imaging of ferritin. M.L.L. gave technical advice.

## Competing interests

The University of California Davis has applied for a patent with L.E.C-L., C.M., and C.T. listed as named inventors (USPTO Serial No: 62/868,790) covering the process for making the cell-free system described in this study. Y.L., M.S., S.L., and M.L.L declare no competing interests.
