## [Peer Review File · Nature Communications]

Reviewers' Comments:

Reviewer #1:

Remarks to the Author:

The manuscript from the group of Professor Cheemeng Tan, entitled, "Holistic engineering of cell-free systems through proteome-reprogramming synthetic circuits," showcases optimized gene expression in cell-free expression systems based upon synthetically reprogrammed expression extracts grouped from "consortia" of 7 and 18 cell strains.

The overall extract has been characterized in detail in an extremely thorough fashion. There's no reason to believe these extracts are not highly functional and the interpretation of the data, including what remains to be done in future studies, is well thought out.

The abstract made no mention of a specific result, outside of a 5-fold increase in expression in "classical cell-free systems." As a result, as I read through this very well-written account of well-designed and executed experiments, I was curious about what the specific deliverable would be showcasing the function of this system.

The authors chose to express ferritin, Cas9, and linear protein segments. I believe all of these choices to be well-made. I was particularly intrigued by ferritin and Cas9, which have immediately obvious applications beyond the characterization of the study. These data were sufficient for me to consider this appropriate for publication, which I recommend.

I would consider amending the abstract to include more specifics on the categories of "proof-of-concepts" that have been included. (In my opinion, using ferritin or Cas9 is unnecessary, but perhaps mentioning "nanocage" and "gene-editing nuclease" may allow this paper's impact to be appreciated by a wider audience from material scientists to molecular biologists.

I do have the following minor comments, which should be addressed:

1. The introduction section appears to include a liberal use of "synthetic circuit," whereas the authors appropriate refer to parts of their work as "synthetic modules." With the exception of true circuits (i.e., a circuitous signaling network where a feedback or feedforward mechanism exists, such as repression or activation), synthetic circuit should be avoided. It would be good if the authors could double-check their usage here.

2. On page 10, please change "in macromolecule synthesis (e.g. chaperones and ribosomal proteins); and" to "in macromolecule synthesis (e.g., chaperones and ribosomal proteins); and" for formatting to be consistent.

3. In the text, you use "ferritin" but in the figure legend, you use "Ferritin." The usage "ferritin" seems more appropriate.

Reviewer #2:

Remarks to the Author:

The authors demonstrate that an E. coli consortium, consisting of seven strains, each over-expressing translational machinery components yields a cell free extract that improves cell free protein synthesis. The authors claim this result is caused by reprogramming of the proteome. The authors further claim that the engineered cell free extract improves the production of difficult to express proteins. Furthermore, with supplementation of a strain over-expressing Gam protein to the consortium, they achieve yields of 75 % using linear templates as compared to those achieved with plasmid DNA.

Microbial consortia have previously been used in preparation of the PURE system. Furthermore, as acknowledged by the authors, several strategies have been used to genomically recode *E. coli* to alter the concentration of various components in the resulting extract. However, to the best of my knowledge using a consortium of cells expressing various translational components to improve cell free protein synthesis has not previously been implemented and therefore represents a novel contribution to the field.

With respect to the authors' claims in the discussion that their approach could be used more generally for achieving other goals in mammalian cells etc., the author's fail to realise that transcriptional rewiring and other so-called "holistic" approaches have previously been used. An example of this is seen by the work by Bayer and colleagues (doi: 10.1093/nar/gkx197).

Several components of the Supplementary Materials, including Figures, Text and a Table were not available, making it impossible to completely evaluate this work. With that said I found their work generally convincing with the exception of the following issues:

- The authors don't compare their CFPS system against the commercial S30 system in semi-continuous exchange mode (Figure 1C). However, this is done in Batch mode (Figure 1B). I suspect this is because a semi-continuous exchange protocol is not available for the commercial, however they should state this or otherwise explain why the commercial system wasn't used in this context.
- It is not clear to me why translation machinery was supplemented into two different strains (BL-EWCE and BL-CFPWCE) to generate Figures 2B & C? Should this not be the same strain to control for this variable?
- How much DNA was added to the "Plasmid Template" vs "Linear Template" sample in Figure 4D? Should this not be controlled for? Otherwise their system could be compared to a commercially available kit for linear templates e.g. Promega *E. coli* S30 Extract System for Linear Templates (L1030).
- The authors have not conducted a two-tail t-test for the data in Figure 4A. However, this has been done for the data in Figure 4C and in Figure 1. Is this result not significant?

The work presented here suggests that the expression of synthetic circuits can improve the performance of cell free protein synthesis systems through the over-expression of selected components and consequential changes to the host. This work could form the basis for further investigations into achieving other goals in the area of cell free protein synthesis. An example could be improving the incorporation of non-natural amino acids into proteins.

The authors provide a detailed Materials and Methods section. However, as previously mentioned a large proportion of the Supplementary Materials was not provided.

Additional comments

- The author's mention that their CFPS systems display a 20-minute lag-period (Figure 1D). However, they don't explain why this might be the case.
- The author's state that high growth rates are associated with the "upregulation of proteins involved in macromolecule synthesis... and downregulation of metabolic proteins". There is no citation for this statement or data to support it. Are metabolic proteins in general not required for high growth?
- Figure 3C y-axis label should be "-log₁₀(p-value)", currently it is "log₁₀(p-value)".

Reviewer #1 (Remarks to the Author):

The manuscript from the group of Professor Cheemeng Tan, entitled, "Holistic engineering of cell-free systems through proteome-reprogramming synthetic circuits," showcases optimized gene expression in cell-free expression systems based upon synthetically reprogrammed expression extracts grouped from "consortia" of 7 and 18 cell strains.

The overall extract has been characterized in detail in an extremely thorough fashion. There's no reason to believe these extracts are not highly functional and the interpretation of the data, including what remains to be done in future studies, is well thought out.

We thank the reviewer for his positive comments regarding our work.

The abstract made no mention of a specific result, outside of a 5-fold increase in expression in "classical cell-free systems." As a result, as I read through this very well-written account of well-designed and executed experiments, I was curious about what the specific deliverable would be showcasing the function of this system.

The authors chose to express ferritin, Cas9, and linear protein segments. I believe all of these choices to be well-made. I was particularly intrigued by ferretin and Cas9, which have immediately obvious applications beyond the characterization of the study. These data were sufficient for me to consider this appropriate for publication, which I recommend.

I would consider amending the abstract to include more specifics on the categories of "proof-of-concepts" that have been included. (In my opinion, using ferritin or Cas9 is unnecessary, but perhaps mentioning "nanocage" and "gene-editing nuclease" may allow this paper's impact to be appreciated by a wider audience from material scientists to molecular biologists.

We thank the reviewer for the suggestions. We have made changes in the abstract, addressing the concerns regarding the categories that we used as a proof of concept.

I do have the following minor comments, which should be addressed:

1. The introduction section appears to include a liberal use of "synthetic circuit," whereas the authors appropriate refer to parts of their work as "synthetic modules." With the exception of true circuits (i.e., a circuitous signaling network where a feedback or feedforward mechanism exists, such as repression or activation), synthetic circuit should be avoided. It would be good if the authors could double-check their usage here.

We have double-checked and modified the use of the term "synthetic circuit" to improve clarity.

2. On page 10, please change "in macromolecule synthesis (e.g. chaperones and ribosomal proteins); and" to "in macromolecule synthesis (e.g., chaperones and ribosomal proteins); and" for formatting to be consistent.

This change has been made in the text.

3. In the text, you use "ferritin" but in the figure legend, you use "Ferritin." The usage "ferritin" seems more appropriate.

We agree that the use of ferritin is more appropriate and have changed all instances of “Ferritin” to “ferritin”.

Reviewer #2 (Remarks to the Author):

The authors demonstrate that an E. coli consortium, consisting of seven strains, each over-expressing translational machinery components yields a cell free extract that improves cell free protein synthesis. The authors claim this result is caused by reprogramming of the proteome. The authors further claim that the engineered cell free extract improves the production of difficult to express proteins. Furthermore, with supplementation of a strain over-expressing Gam protein to the consortium, they achieve yields of 75 % using linear templates as compared to those achieved with plasmid DNA.

Microbial consortia have previously been used in preparation of the PURE system. Furthermore, as acknowledged by the authors, several strategies have been used to genomically recode E. coli to alter the concentration of various components in the resulting extract. However, to the best of my knowledge using a consortium of cells expressing various translational components to improve cell free protein synthesis has not previously be implemented and therefore represents a novel contribution to the field.

We thank the reviewer for the comments and for acknowledging the novelty of our work.

With respect to the authors claims in the discussion that their approach could be used more generally for achieving other goals in mammalian cells etc., the author’s fail to realise that transcriptional rewiring and other so-called “holistic” approaches have previously been used. An example of this is seen by the work by Bayer and colleagues (doi: 10.1093/nar/gkx197).

We thank the reviewer for pointing out this study. We have added a sentence in the discussion (Page 17) about transcriptional rewiring acknowledging the similarity of the approach and pointed out a couple of studies that use this approach for the production of proteins and metabolites.

Several components of the Supplementary Materials, including Figures, Text and a Table were not available, making it impossible to completely evaluate this work. With that said I found their work generally convincing with the exception of the following issues:

We thank the reviewer for bringing this to our attention. We have now included the Supplementary Materials in full.

- The authors don’t compare their CFPS system against the commercial S30 system in semi-continuous exchange mode (Figure 1C). However, this is done in Batch mode (Figure 1B). I suspect this is because a semi-continuous exchange protocol is not available for the commercial,

however they should state this or otherwise explain why the commercial system wasn't used in this context.

We thank the reviewer for this comment. The continuous exchange reaction with the S30 system was conducted based on the protocol used for the other extracts. We found the S30 system to perform far worse than our system but felt that it was an unfaithful representation of the system because the protocol was not defined by the manufacturer. We added the explanation to the text (Page 4).

- It is not clear to me why translation machinery was supplemented into two different strains (BL-EWCE and BL-CFPWCE) to generate Figures 2B & C? Should this not be the same strain to control for this variable?

We thank the reviewer for this question. This set of independent experiments are intended to rule out two possible explanations for the increase in cell-free protein synthesis besides the potential auxiliary effect that creates a more favorable environment for CFPS. First, with the addition of purified translation machinery to BL-E_{WCE} we show that an increased concentration of protein machinery is not the only responsible for the increased protein expression shown by our multi strain CFPS systems. Second, we performed the addition of purified translation machinery proteins to BL-CFP_{WCE} to rule out any additional beneficial effect that plasmid maintenance and protein overexpression (CFP) may be causing in the whole cell lysate.

We acknowledge that the current labeling of figures may be misleading and give a false impression of the objective of experiments in Fig 2. We have modified the labeling of the figures, figure captions, and text to better explain the intention of these experiments and avoid any possible confusion. The detailed experimental setup can be found in Materials and Methods, Sections M.3. and M.6.

- How much DNA was added to the “Plasmid Template” vs “Linear Template” sample in Figure 4D? Should this not be controlled for? Otherwise their system could be compared to a commercially available kit for linear templates e.g. Promega E. coli S30 Extract System for Linear Templates (L1030).

We thank the reviewer for this question. The intention of this experiment was not to make a comparison with commercially available linear template expression systems but to demonstrate the modularity of the use of a bacterial consortium for the production of cell-free lysates. This modularity can be exploited to custom modify the whole cell lysates according to the requirements of a given experiment. The reactions that used plasmid DNA as a template were assembled using 100 ng (per 10 uL reaction) as specified in Materials and Methods, section M.3. *Assembly of CFPS reactions for batch experiments.* The reactions performed using linear template were assembled by directly using the product of the amplification of 1 ng of plasmid DNA using the TempliPhi rolling circle amplification kit as specified in section Materials and Methods, M.3. *Assembly of CFPS reactions using linear template.* The total amount of DNA added to each reaction was not quantified before addition to CFPS reactions assembled with BL-7S or BL-Gam. However, the same isothermal amplification reactions were used for supplementing linear template to both reactions to guarantee an unbiased result. The precise

concentration of DNA obtained through RCA could not be quantified using a Nanodrop spectrophotometer. This issue is because even in the absence of input DNA, the RCA reaction yields nonspecific products. However, according to the manufacturer's indications, we estimate that the amount of double-stranded linear DNA added to each CFPS reaction is between 150-500 ng.

We acknowledge that the experimental setup can give rise to confusion. We have added more information regarding the assembly of CFPS reactions using the linear template in Materials and Methods, Section M.3. We have also pointed out that further optimization and benchmarking of our system should be carried out in future studies to demonstrate the improvement over commercial linear template CFPS systems (Page 15).

- The authors have not conducted a two-tail t-test for the data in Figure 4A. However, this has been done for the data in Figure 4C and in Figure 1. Is this result not significant?

Thanks for bringing this to our attention. We have conducted a two-tail t-test for the data in Figure 4A. We have included this information in the chart and the figure caption.

The work presented here suggests that the expression of synthetic circuits can improve the performance of cell free protein synthesis systems through the over-expression of selected components and consequential changes to the host. This work could form the basis for further investigations into achieving other goals in the area of cell free protein synthesis. An example could be improving the incorporation of non-natural amino acids into proteins.

We thank the reviewer for the comment. We have included a sentence in the discussion and a few references illustrating potential applications of our work to other research areas in CFPS (Page 17).

The authors provide a detailed Materials and Methods section. However, as previously mentioned a large proportion of the Supplementary Materials was not provided.

Thanks for bringing this to our attention. We have now included the Supplementary Materials in full.

Additional comments

- The author's mention that their CFPS systems display a 20-minute lag-period (Figure 1D). However, they don't explain why this might be the case.

We thank the reviewer for this comment. During this 20-minute lag period, T7 RNAP starts and sustains mRNA synthesis until the mRNA reaches the concentration necessary for starting protein synthesis. We have added the explanation in the main text (Pages 4&5).

- The author's state that high growth rates are associated with the "upregulation of proteins involved in macromolecule synthesis... and downregulation of metabolic proteins". There is no citation for this statement or data to support it. Are metabolic proteins in general not required for high growth?

Thanks for bringing this to our attention. We have added the appropriate literature references (references 4 and 29, Page 11) to support our statement.

- Figure 3C y-axis label should be “ $-\log_{10}(\text{p-value})$ ”, currently it is “ $\log_{10}(\text{p-value})$ ”.

We thank the reviewer for this comment. Figure 3C has been modified to incorporate this change.

Reviewers' Comments:

Reviewer #2:

Remarks to the Author:

The authors have addressed all of my comments satisfactorily.